# Achievement Goals across Persistence—Validation of the Spanish Version of the Motivational Persistence Scale

**DOI:** 10.3390/ijerph19148474

**Published:** 2022-07-11

**Authors:** Iñaki Quintana, Maria Isabel Barriopedro, Luis Miguel Ruiz Perez

**Affiliations:** 1Faculty of Health Sciences, Universidad Francisco de Vitoria (UFV), 28223 Madrid, Spain; 2Department of Social Sciences of Physical Activity and Sport, Universidad Politécnica de Madrid (UPM), 28040 Madrid, Spain; misabel.barriopedro@upm.es (M.I.B.); luismiguel.ruiz@upm.es (L.M.R.P.)

**Keywords:** persistence, validation, reliability, deliberate practice, performance

## Abstract

Background: An important aspect of achievement goals is the persistence and determination that the person possesses in order to achieve it. Spain does not have an adequate instrument for its measurement. First, this article had the aim of adapt and validate the Motivational Persistence Scale of Constantin et al. in a Spanish population and athletes. Second, it had the aim of prove the relationship with deliberate practice and performance. Methods: In this study, 384 university students participated, where the factor structure was analyzed by means of a Confirmatory Factor Analysis (CFA). In study 2 of 169 athletes was used to confirm its validity in a homogeneous population and its predictive capacity on the hours of deliberate practice (DP) and performance. Results: The AFC showed a two-factor structure, reducing the original three-factor structure, presenting a good fit in both Spanish and homogeneous population of athletes and achieving a significant predictive capacity on deliberate practice. The new dimensions were Purpose Pursuing (PP) and Recurrence of Unattained Purposes (RUP). Conclusions: Overall, our results provide evidence that this scale could be a useful tool for the assessment of Persistence in the Spanish adult and athlete population.

## 1. Introduction

A relevant aspect for achievement goals is that the person persists in their determination to achieve them, that they do not give up, that they have the necessary motivation to remain firm and to continue in spite of the possible difficulties they may encounter.

Persistence has been defined as the amount of time spent on difficult issues and also by the reluctance to settle for an easier outcome [1]. There are numerous concepts that along persistence have highlighted the insistence of individuals to achieve a goal. Concepts such as perseverance [2], grit [3], or tenacity [4]. All of them emphasize the intensity with individuals are involved in the objectives they set, being for some a mere expression of their character [5]. Constantin et al. [5] consider that persistence should be differentiated according to whether the goal is close and short-term or whether it is a goal with a longer time perspective, i.e., long-term, since this circumstance could affect the level of effort to be invested. When short-term goals are pursued, the individual concentrates on maintaining his or her attentional effort and withstanding boredom, stress and even frustrations, distractions or inconveniences that may arise. On the other hand, when the individual pursues long-term goals, the ability to remain committed to their achievement is demanded, considering necessarily amounts of resources, as well as a prolonged investment of time [5].

Persistence is strongly connected with academic achievement, physical education [6,7], foreign language learning and sports performance [8,9]. Particularly, in the literature, this scale has been mainly studied in student populations [10,11,12,13]. In the study of Erte and Ari [11] students with high motivational persistence to pursue long-term purposes and pursue current purposes have less tendency to procrastinate. In sports context, we couldn’t found studies using this scale. However, different studies have analyzed persistence by relating it to sport dropout [13] or intrinsic motivation [14,15,16]. Thus, for example, in the study by Jõesaar et al. [17], more persistent soccer players presented higher levels of intrinsic motivation. In addition, persistence has been found to be highly related to self-efficacy beliefs in students [18] and in tennis players, as the case of Halper and Vancouver’s study [19]. The results from the current study confirm that self-efficacy can foster persistence when one is aware of one’s current state of performance. However, consistent with a control theory view of self-regulation, self-efficacy was unrelated to persistence when feedback was ambiguous. The results have implications for understanding the role of self-efficacy in sports and highlight the importance of replications with extensions. On the other hand, persistence in sport is affected by the role of different social agents [20]. Parents, coaches and peers can influence the athlete to persist in achieving his or her goals. For example, athletes who have been significantly supported for their parents, they have persisted more in their sports [20]. The coach also plays a very important role in the athlete persistence [21,22,23]. This influence can be positive and negative: Positive because the coach can get adherence to athlete training, and negative because they can offer negative feedback to the athletes, reduce their commitment training [24,25].

Another important factor for improve to the performance level and persistence sport is Deliberate Practice (DP) [26]. This concept is based on the fact that those who have overcome great challenges have needed thousands of hours of training for their achievement, which requires high levels of persistence to withstand the difficulties, not to give up, and to return to resume those goals that may have remained unachieved [26]. In the scientific literature, DP is related to increased athlete performance, greater commitment and consequently reaching higher goals [27].

Due to the lack of an instrument in the Spanish version that measures persistence across achieved goals, the aim of this study was to adapt and validate the Motivational Persistence Scale developed by Constantin et al. [5] a Spanish population. a secondary objective, we have tried to focus in the importance of persistence in achievement goals in relation to DP in sports, specifically tennis. We hypothesize that this instrument can be used in the Spanish population to measure motivational persistence and it can help to the scientific community to know the psychometric properties of this scale in other languages.

### Study 1

This study had a correlational, non-experimental and cross-sectional design. To obtain evidence on the validity of the scale, we set out to analyze the validity of the factorial structure, to analyze its reliability, to evaluate its relationship with the Grit scale (convergent validity) and, finally, to analyze the measurement invariance according to sex and age.

## 2. Materials and Methods

### 2.1. Participants

The aleatory simple sample of this study consisted of 384 participants, aged between 13 and 46 years, of whom 297 were males (M = 21; SD = 3.7 years) and 87 females (M = 19.7; SD = 3.4 years). A total of 84.9% were university students in the field of sports science and 15.1% were athletes.

### 2.2. Measures

#### 2.2.1. Motivational Persistence Scale by Constantin et al.

This scale was translated from English to Spanish. For this translation, the recommendations for the adaptation of tests from one culture to another were taken into account [28]. A bilingual translator, with knowledge of the specific contents of the present study and principles of test construction, translated all the items of the English version. A second bilingual translator translated the Spanish version back into English. Finally, two expert researchers in the field compared this translation with the original English scale. Finally, the translators and the researchers agreed on a final version (Table 1).

The Persistence scale, consisting of 13 items, evaluates three dimensions: (a) Current purposes pursuing (4 items), (b) Long-term purposes pursuing (4 items) and (c) Recurrence of unattained purposes (5 items).

#### 2.2.2. Grit-S Scale

The Grit-S Scale of Duckworth et al. [3] was applied in its Spanish version of Barriopedro et al. [29] to assess convergent validity. This instrument presents two factors: Consistency of interests and Perseverance in effort. Its objective is to explore aspects of personality related to passion and courage in the achievement of goals.

### 2.3. Procedure

The participants were invited to collaborate in this studyand they were informed of the anonymity of data and they signed a consent form. The all of them completed, without time limit, the two scales, the Motivational Persistence and the Grit-S scale. Each item of the Motivational Persistence scale was answered on a 5-point Likert-type scale, with 1 (strongly disagree) and 5 (strongly agree). As for the Grit-S scale, each item was answered using a polytomous response format with 5 alternatives: (a) very much like me, (b) mostly like me, (c) somewhat like me, (d) not much like me, and (e) not like me at all.

### 2.4. Data Analysis

The factor structure proposed by Constantin et al. [5] was subjected to a Confirmatory Factor Analysis (CFA) using the Maximum Likelihood (ML) method to estimate the parameters. The following indices were used to evaluate the goodness of fit of the data to the proposed model: Root Mean Squared Error of Approximation (RMSEA) and its 90% confidence interval, Bentler’s Comparative Fit Index (CFI), Goodness of Fit Index (GFI) and the relative Chi-Square value (χ^2^/gl).

In the literature, values of RMSEA ≤ 0.05 or 0.08 are considered as indicators of a good fit or an acceptable fit, respectively; values of CFI ≥ 0.95 and of GFI ≥ 0.95 have been accepted as indicators of a good fit; finally relative Chi-Square values between 2 and 5 are considered as indicators of acceptable fit. To test whether the factor structure was the same in different age groups (under 20 vs. over 20 and males and females), multigroup CFAs were performed. Factor invariance was assessed in a stepwise manner [30,31]. The factorial invariance was assessed progressively: configural invariance (the pattern of factor loadings is the same); metric or weak invariance (in addition to the pattern of loadings, the factorial weights are equal); scalar or strong invariance (in addition to metric invariance, it assumes that the intercepts are equal) and strict invariance (in addition to scalar invariance, it assumes equal variances for the errors).

Because of the difference between Chi-Square tests is very restrictive, the Akaike Information Criterion (AIC) and the comparison of CFI values were also used to compare the fit of the models. If the difference between the CFI values of two nested models is higher than 0.01 in favor of the less restrictive model, the more restrictive model should be rejected [32]. Reliability of the scales was assessed by the α coefficient and item homogeneity from the correlation between the item and the total scale score once the item was removed. In addition, to complement cronbach’s alpha, the omega test (ϖ) was performed [33]. T-tests were used for comparison of factor scores and Cohen’s d was used as an index of effect size. The values 0.2, 0.5 and 0.8 were used as reference values to consider the effect size as small, medium and large, respectively [34]. The analyses were performed using the PASW 21.0 program and the AMOS 21.0 program.

## 3. Results

### 3.1. Dimensionality and Reliability of the Scale

In the first place, the descriptives of each item are presented by mean and standard deviation in Table 2. The estimation of the correlation coefficients between the items was analysed and presented in Table 3.

The three-factor structure proposed by Constantin et al. [5] yielded a similar fit to that reported by the authors: RMSEA = 0.058 (0.046−0.071); CFI = 0.93; GFI = 0.95; χ^2^/139 = 2.3. Only the LLPP factor and the CPP factor showed a significant correlation; however, the estimation of this correlation resulted in a value higher than 1 (r = 1.01). In the original study, the correlation found between these two dimensions was 0.92, indicating that these two factors are assessing a single dimension.

When the internal consistency was analyzed, joining the items corresponding to LLP and CPP dimensions, good values was obtained of 0.788, which improved with the elimination of item 3 to 0.793 (ϖ = 0.796) for this new Purposes Pursuing (PP) dimension, and a value of 0.74 for the RUP dimension, which improved to 0.76 (ϖ = 0.763) with the elimination of item 13. The consistency of the total scale was 0.69 (ϖ = 0.875). The mean correlations of the items with the total scale with the item removed was 0.52 for the PP dimension, 0.56 for the RUP dimension, and 0.34 for the entire scale. After eliminating the items that resulted in an increase in internal consistency (Figure 1), the new two-dimensional scale, presented adequate fit indices: RMSEA = 0.043 (0.025–0.059); CFI = 0.97; GFI = 0.96; χ^2^/53 = 1.71. The items were good indicators of their respective latent factors, with factor loadings between 0.48 and 0.94 for the RUP subscale and between 0.51 and 0.67 for the PP subscale. All weights were statistically significant (*p* < 0.001). Scores on both subscales were independent (r = −0.11; *p* = 0.080).

### 3.2. Convergent Validity

The Grit-S questionnaire was used to assess convergent validity. Considering the original 3-factor Motivational Persistence scale, the Consistency factor of the Grit-S showed significant correlations with the CPP (r = 0.33) and LTPP (r = 0.37) factors, and an inverse relationship with the RUP factor (r = −0.29). The Perseverance factor only showed significant correlations with the CPP (r = 0.56) and LTPP (r = 0.49) factors (Table 4).

Considering the new two-factor scale, both the Consistency factor and the Perseverance factor of the Grit-S showed significant correlations with the PP factor (r = 0.38 and r = 0.56, respectively) and the Consistency factor of Grit-S showed an inverse relationship with the RUP factor (r = −0.31).

### 3.3. Factorial Invariance

In order to test for factorial invariance according to age, two groups of subjects were formed, those aged 20 years or less (N = 215) and those above this age (N = 169). The scale showed acceptable fit indices (Table 5) both for the group of subjects aged 20 years or less (RMSEA = 0.044 and CFI = 0.97) and for the group of subjects over 20 years (RMSEA = 0.049 and CFI = 0.96). The indices obtained allow us to accept the equivalence of the pattern of factor loadings (configural invariance) for both groups of subjects. Although the Chi-square value was significant, the rest of the indices supported this conclusion (RMSEA = 0.033 and CFI = 0.966). When restrictions on factor weights (metric invariance) were added to the model, a good model fit was obtained (RMSEA = 0.035 and CFI = 0.957). On the other hand, Akaike’s information criterion (AIC = 257.6) and Bentler’s comparative index did not undergo appreciable modifications. These results allow us to accept the metric invariance.

When the intercept restriction (scalar invariance) was added to the model, Bentler fit index (CFI = 0.952) and root mean squared error of approximation (RMSEA = 0.035) provided acceptable fit values, the CFI reduction was less than 0.01 with respect to the metric invariance model and the AIC (251.3) did not undergo a notable variation, data that lead us to accept scalar invariance. The model assuming strict invariance also showed acceptable fit indices, with no significant change in CFI (reduced by 0.006) or AIC (246.5) with respect to the scalar model, so that strict invariance can be accepted.

The scale showed acceptable fit indices (Table 5) for males (RMSEA = 0.048; CFI = 0.96) and females (RMSEA = 0.00; CFI = 0.99). The indices obtained allow us to accept the equivalence of the pattern of factor loadings (configural invariance) for both groups of subjects (RMSEA = 0.027 and CFI = 0.977). When restrictions on factorial weights (metric invariance) were added to the model, a good model fit was obtained (RMSEA = 0.026 and CFI = 0.975). On the other hand, Akaike’s information criterion (AIC = 238.5) and Bentler’s comparative index (CFI = 0.975) had not appreciable modifications. These results allow us to accept the metric invariance.

When the intercept constraint (scalar invariance) was added to the model, both the Bentler fit index (CFI = 0.964) and the root mean squared error of approximation (RMSEA = 0.030) provided acceptable fit values, although the CFI reduction was 0.011 with respect to the metric invariance model. After eliminating the equality of intercepts in item 8, we obtained indices that lead us to accept partial scalar invariance. The model assuming strict invariance also showed acceptable fit indices, but the CFI reduction was 0.014 with respect to the scalar model. After eliminating the equality of intercepts and error variances of item 8, in addition to acceptable fit indices, the CFI variance was 0.01 and the Akaike information criterion was virtually unchanged.

### 3.4. Study 2

This second study had a correlational, non-experimental and cross-sectional design. The aims of this second study were, on the one hand, to confirm the 2-factor structure for the Persistence Scale, from the previous study, in a sample of tennis players and, on the other hand, to evaluate the ability of these two factors to predict the hours of deliberate practice (DP).

## 4. Materials and Method

### 4.1. Participants

In this study 169 tennis players participated across 14 different clubs. This sample was selected for convenience The age was between 14 and 18 years, of whom 44 were males (M = 15.8; SD = 1 years) and 125 females (M = 15.9; SD = 1.1 years). Among the males, 75% were cadets and 25% juniors; 59.1% had competed at the regional level and 40.9% at the national level. Of the female tennis players, 69.6% were cadets and 30.4% juniors; 59.2% had competed at the regional level and 40.8% at the national level. After parents of players had read and signed the informed consent form approved by the university’s Institutional Board, they were able to continue to the questionnaires, if they consented.

### 4.2. Measures

The participants completed a questionnaire with sociodemographic questions (sex and age) and questions related to their sports activity. The calculation of DP hours was made from the response to two questions: (1) number of hours dedicated each day to activities aimed at improving their sports performance, and (2) days per week of training in a typical week of the season. These measures can be considered as an acceptable approximation to the amount of weekly DP [35].

The athletes also reported their performance (national ranking) and their competitive level (regional or national) according to the number of tournaments they played. In addition, they were classified into different categories, cadet or junior according to their year of birth. All subjects voluntarily completed the Motivational Persistence Scale [5] from the previous study.

### 4.3. Procedure

The participants were invited to collaborate in this study, and they were informed of the anonymity of data and they signed a consent form. The participants were invited to participate in the study through an e-mail sent by the director of the tennis club to which they belonged. The players, after their usual training, completed a booklet included the Motivational Persistence Scale and some questions about their sporting profile. They had no time limit.

### 4.4. Data Analysis

The factor structure found in the previous study was subjected to a CFA in this sample of athletes. The evaluation of the ability of the factors of this scale to predict DP hours and performance was performed using structural equation modeling. The Robust Maximum Likelihood method [36] was used to correct the goodness-of-fit indices (RMSEA and its 90% confidence interval, CFI and Chi-Square value (χ^2^)). Reliability of the scales was assessed by the α coefficient and item homogeneity from the correlation between the item and the total scale score once the item was removed. In addition, to complement the cronbach alpha information, the omega test (ϖ) was performed [33]. Two multivariate analyses of covariance (one for cadets and one for juniors), using age as a covariate, were performed to assess whether groups of different skill level (those with a ranking of 1500th place or better versus those ranked 5000th place or worse) and sex differed on persistence factors. Post hoc comparisons were performed using the Bonferroni test. As an index of effect size, partial Eta squared (η^2^p) was calculated. Values up to 0.06, from 0.06 to 0.14 and ≥0.14 will be interpreted as small, medium and large effect sizes, respectively [34]. The risk level was set at 0.05 for all analyses which were performed using SPSS version 21 (IBM Corp., Armonk, NY, USA) and the EQS program. 

## 5. Results

### 5.1. Persistence Scale, Deliberate Practice and Sport Performance

The two-dimensional Persistence scale, PP and RUP, of 7 and 4 items respectively, presented adequate fit indices: RMSEA = 0.018 (0.000–0.056); CFI = 0.992; χ^2^ S-B (43) = 45.34, *p* = 0.3746. The items were good indicators of their respective latent factors, with factor loadings between 0.43 and 0.71 for the RUP subscale and between 0.50 and 0.72 for the PP subscale (Figure 2). All weights were statistically significant (*p* < 0.001). Scores on both subscales were independent (r = −0.06; *p* = 0.541). When internal consistency was analyzed, a value of 0.78 (ϖ = 0.771) was obtained for PP subscale, and a value of 0.65 (ϖ = 0.65) for the RUP dimension. The consistency of the total scale was 0.65 (ϖ = 0.749). The correlations of the items with the total scale with the item removed were greater than 0.40 except for item 1 which was 0.35. The SEM model for predicting deliberate practice hours (Figure 3) showed a good fit (RMSEA = 0.031 [90% CI: 0.000–0.060]; CFI = 0.975; χ^2^ S-B (53) = 61.38, *p* = 0.200). The weight of the subscale RUP subscale scores on practice hours was statistically significant (β = −0.22, *p* = 0.019) but not that of the PP subscale (β = 0.16, *p* = 0.069).

The SEM model for predicting the classification of tennis players from the persistence scale (Figure 4) showed a good fit (RMSEA = 0.026 [90% CI: 0.000–0.057]; CFI = 0.981; χ^2^ S-B (53) = 58.85, *p* = 0.270). The weight of the RUP subscale scores on practice hours was statistically significant (β = 0.23, *p* = 0.019) but not that of the PP subscale (β = −0.09, *p* = 0.293).

### 5.2. Motivational Persistence and Performance Levels

Descriptives for weekly DP hours and factor scores on the Persistence scale are presented in Table 6. For cadet tennis players there was multivariate effect of competitive level (F3, 75 = 14.03, *p* < 0.001, η^2^p = 0.360). The covariate age (F3, 75 = 0.71, *p* = 0.552), sex (F3, 75 = 1.98, *p* = 0.124, η^2^p = 0.073), the interaction between sex, and competitive level (F3, 75 = 0.75, *p* = 0.529, η^2^p = 0.029) had not significant multivariate effects. Univariate analysis showed that scores on the RUP scale were not different among subjects ranked below rank 5000 than among those ranked above 1500 (F1, 77 = 3.86, *p* = 0.053, η^2^p = 0.048). Conversely, subjects ranked below rank 5000 scored significantly lower on the PP scale than those ranked above 1500 (F1, 77 = 5.01, *p* = 0.028, η^2^p = 0.061). PD hours were significantly higher among those ranked above 1500 (F1, 77 = 36.70, *p* < 0.001, η^2^p = 0.323). After eliminating females from the analysis, due to small sample size, for junior tennis players the multivariate analysis yielded significant effect of competitive level (F3, 18 = 7.54, *p* = 0.002, η^2^p = 0.557). The effect of the covariate age was not significant (F3, 18 = 1.34, *p* = 0.292, η^2^p = 0.183). Only in PD hours was there a univariate effect of competitive level (F1, 20 = 24.50, *p* < 0.001, η^2^p = 0.360), with a higher PD time for those ranked below 1500. There was no univariate effect of competitive level on either RUP scores (F1, 20 = 0.74, *p* = 0.400, η^2^p = 0.036) or PP scores (F1, 20 = 0.19; *p* = 0.672, η^2^p = 0.009).

## 6. Discussion

The purpose of the present study was to contrast the importance of persistence in the achievement goals for DP in sports performance. On the one hand, in the first validation study in the Spanish population, the analysis of the factorial validity of the scores through confirmatory factor analysis, yielded a similar fit to that offered by the authors, however, the fact of finding an impossible correlation value between the factors LTPP and CPP, caused the items of these two factors to be grouped into a single dimension. This possibility should have been highlighted by the original study of Constantin et al. [5] because they found an estimating correlation of 0.92 between them. However, the authors maintained a three-factor structure. Our results are also not in line with the Turkish validation [37]. In this validation across university students, the authors maintained a 3-factor model, however, the results of the confirmatory analysis were similar to this research. Regarding the reliability of this Spanish version of the two-factor Motivational Persistence Scale, individual factor and total scale obtained acceptable internal consistency. This result was similar to Constantin et al. [5] and Sariçam et al. [37]. In addition, as expected, the Consistency of interest and Perseverance in effort factor scores of the Grit-S scale correlated positively and significantly with the Short-Term and Long-Term purposes pursuing factors of the Motivational Persistence scale.

These results replicated those already reported by Constantin et al. [5]. Unifying the two dimensions implies assuming that reaching goals is a global aspect beyond their temporal dimension, which does not coincide with what was proposed by Trope and Liberman [38,39], who considered that the proximity or remoteness of the goals activated different associations and different self-regulatory mechanisms. For the participants in this study, this differentiation did not exist. Regarding the validation of the Motivational Persistence scale in a homogeneous sample of athletes, the results showed good fit indices. The items of the two dimensions were good indicators of their latent factors. The model showed a good fit when we analyzed the predictive validity of this scale on DP hours and ranking of participants. RUP dimension was able to significantly predict both variables. According to the study of reference [5], this dimension indicates that the individual does not set aside purposes that he had it but he did not achieve, returning to them in order to reach them. These goals can coexist with more current goals at the same time. For tennis players who wants to improve, it´s an obligation increase his practice and improve his ranking, achieving all the goals, and not forgetting the ones they did not achieve. Players who do not forget old projects, they accumulate more DP and, consequently, have better ranking. One of the factors that could have influenced in players persistence may be age. Onturk and Yildiz [12] concluded in their study in sport science students that older students (over 25 years) were more persistent, unlike our players whose were younger However, to date there are no studies that allow comparative analyzes relating deliberate practice and the persistence scale in a sports context. Our results suggest that motivation to engage in deliberate practice not only contains elements of the will to improve performance, but also of the will to attain exceptional levels of performance [40]. In addition any studies have been found in spanish language with this structure.

Studies have analyzed the relationships between this scale of Motivational Persistence in professional contexts. Demir and Doner [10] found a positive correlation between RUP and PP with job performance among candidates for teachers. However, in the study of Tutu and Constantin [41] found no correlation between them. Nevertheless, it can be affirmed with the present work, that Motivational Persistence is related to sport improvement [26].

## 7. Conclusions

The Spanish version of the Motivational Persistence scale modifies the original scale structure of the Constantin et al. [5]. The resulting scale presents a good reliability of the total scale as well as of its two dimensions. In addition, the two-factor Motivational Persistence scale yields significant predictive ability on deliberate practice hours and ranking in a homogeneous sample of athletes. The resulting scale may be a good instrument to analyze persistence in goal attainment in Spanish-speaking environments. It is recommended to continue with the study of this scale in other populations and levels of expertise.

## Figures and Tables

**Figure 1 ijerph-19-08474-f001:**
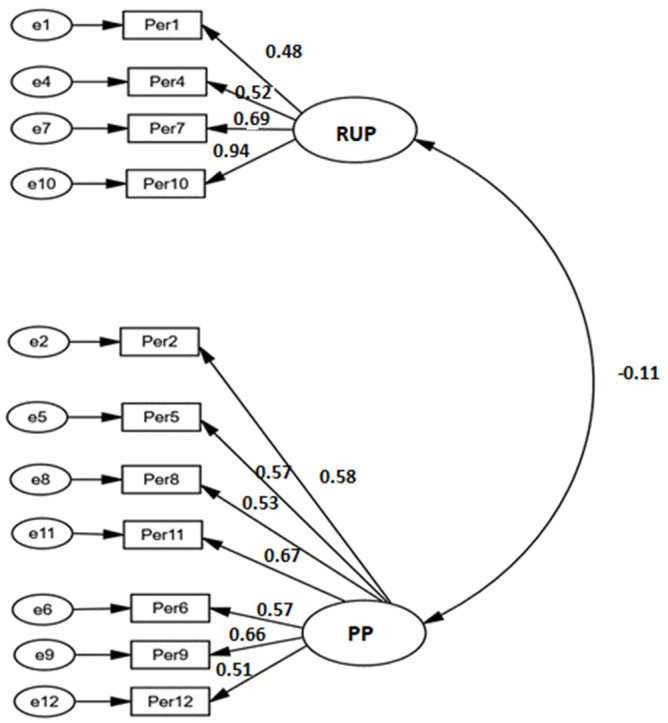
Structural model for the Persistence scale. Standardized weights (RUP: Recurrence of unattained purposes; PP: Purposes Pursuing).

**Figure 2 ijerph-19-08474-f002:**
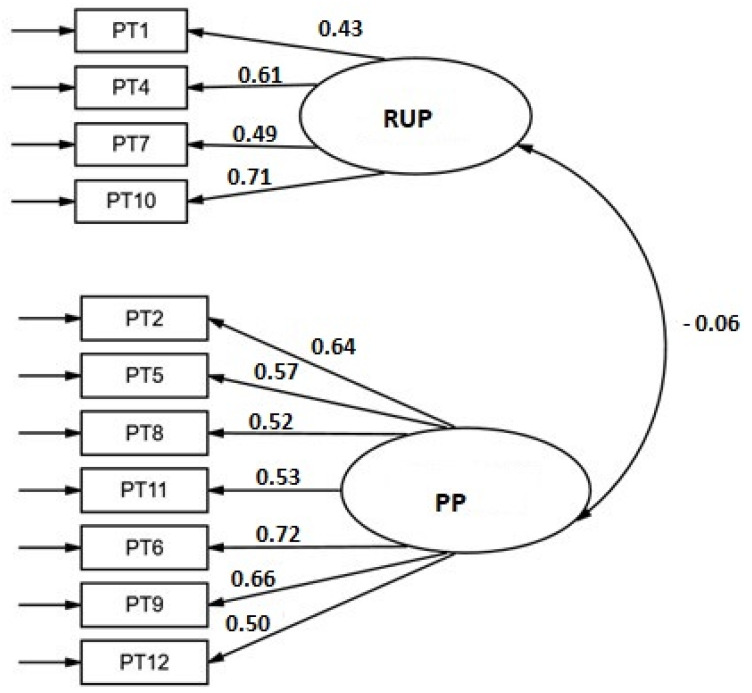
Structural model for the Persistence scale. Standardized weights.

**Figure 3 ijerph-19-08474-f003:**
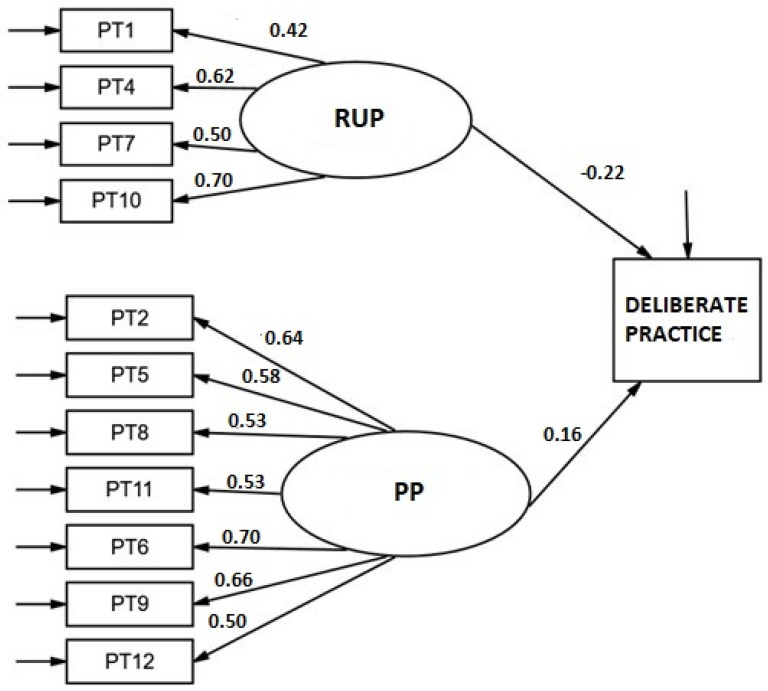
Structural model for the relationship between the latent variables of the Persistence scale and hours of Deliberate Practice. Standardized weights.

**Figure 4 ijerph-19-08474-f004:**
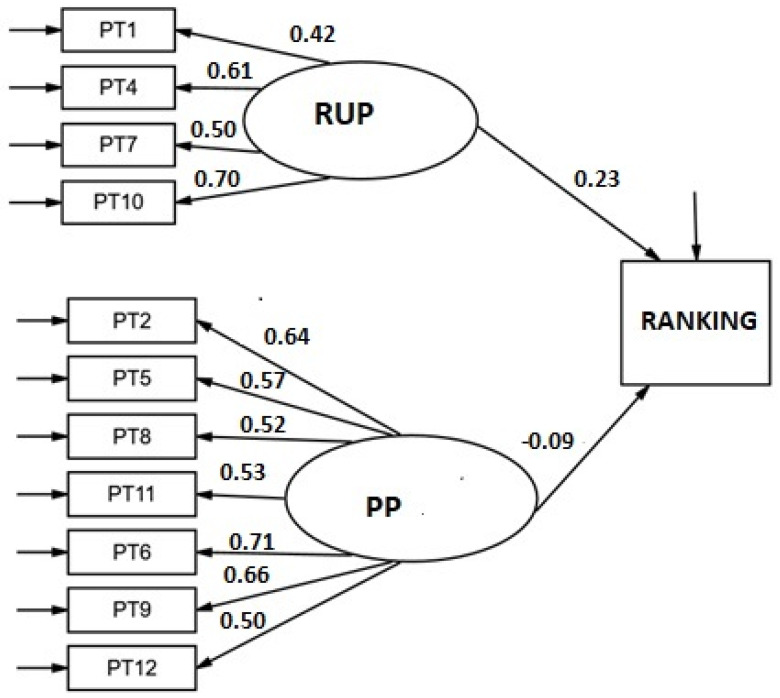
Structural model for the relationship between the latent variables of the Persistence scale and ranking. Standardized weights.

**Table 1 ijerph-19-08474-t001:** List of the items of the Motivational Persistence scale, with indication of the dimension to which each item belongs.

1. I often come up with new ideas on an older problem or project.	RUP
2. I remain motivated even in activities that spread in several months.	LTPP
3. I have a good capacity to focus on daily tasks.	CPP
4. From time to time I imagine ways to use opportunities that I have given up.	RUP
5. Long term purposes motivate me to surmount day to day difficulties.	LTPP
6. Once I decide to do something, I am like a bulldog: I don’t give up until I reach the goal.	CPP
7. Even though it doesn’t matter anymore, I keep thinking of personal aims that I had to give up.	RUP
8. I purposefully pursue the achievement of the projects that I believe in.	LTPP
9. I continue with a difficult task even when others have given up on it.	CPP
10. I often find myself thinking about older initiatives that I had abandoned.	RUP
11. I keep on investing time and effort in ideas and projects that require years of work and patience.	LTPP
12. The more difficult a task is, the more determined I am to finish it.	CPP
13. It’s hard for me to detach from an important project that I had given up in favor of others.	RUP

RUP: Recurrence of Unattained Purposes; LTPP: Long-Term Purposes Pursuing; CPP: Current Purposes Pursuing).

**Table 2 ijerph-19-08474-t002:** Descriptive statistics for Motivational Persistence items.

	Mean	SD
**Per2**	3.92	0.848
**Per5**	3.89	0.866
**Per6**	3.89	0.910
**Per8**	4.08	0.735
**Per9**	3.9	0.826
**Per11**	3.48	0.985
**Per12**	3.85	0.964
**Per1**	3.14	0.947
**Per3**	3.65	0.969
**Per4**	3.14	0.954
**Per7**	2.98	1.043
**Per10**	2.72	1.037
**Per13**	3.56	0.946

SD: Standard Desviation.

**Table 3 ijerph-19-08474-t003:** Pearson correlation among items of Motivational Persistence scale.

	Per2	Per5	Per6	Per8	Per9	Per11	Per12	Per1	Per3	Per4	Per7	Per10	Per13
**Per2**	1	0.400	0.391	0.365	0.310	0.382	0.254	−0.058	0.376	−0.003	−0.102	−0.094	0.091
		<0.001	<0.001	<0.001	<0.001	<0.001	<0.001	0.260	<0.001	0.961	0.045	0.067	0.076
**Per5**	0.400	1	0.328	0.358	0.328	0.361	0.324	0.090	0.160	0.124	0.009	−0.025	0.133
	<0.001		<0.001	<0.001	<0.001	<0.001	<0.001	0.078	0.002	0.015	0.867	0.630	0.009
**Per6**	0.391	0.328	1	0.337	0.364	0.373	0.288	−0.084	0.271	0.025	−0.066	−0.101	0.124
	<0.001	<0.001		<0.001	<0.001	<0.001	<0.001	0.100	<0.001	0.630	0.196	0.048	0.015
**Per8**	0.365	0.358	0.337	1	0.476	0.408	0.292	0.022	0.162	0.056	−0.056	−0.061	0.198
	<0.001	<0.001	<0.001		<0.001	<0.001	<0.001	0.668	0.001	0.277	0.277	0.234	<0.001
**Per9**	0.310	0.328	0.364	0.476	1	0.461	0.363	0.058	0.199	0.070	−0.066	−0.072	0.140
	<0.001	<0.001	<0.001	<0.001		<0.001	<0.001	0.259	<0.001	0.169	0.195	0.161	0.006
**Per11**	0.382	0.361	0.373	0.408	0.461	1	0.360	−0.015	0.260	−0.002	−0.045	−0.104	0.069
	<0.001	<0.001	<0.001	<0.001	<0.001		<0.001	0.770	<0.001	0.962	0.380	0.041	0.179
**Per12**	0.254	0.324	0.288	0.292	0.363	0.360	1	0.063	0.123	0.031	−0.016	−0.018	0.096
	<0.001	<0.001	<0.001	<0.001	<0.001	<0.001		0.218	0.016	0.546	0.749	0.725	0.060
**Per1**	−0.058	0.090	−0.084	0.022	0.058	−0.015	0.063	1	−0.081	0.377	0.265	0.451	0.123
	0.260	0.078	0.100	0.668	0.259	0.770	0.218		0.112	<0.001	<0.001	<0.001	0.016
**Per3**	0.376	0.160	0.271	0.162	0.199	0.260	0.123	−0.081	1	0.082	0.046	−0.025	0.084
	<0.001	0.002	<0.001	0.001	<0.001	<0.001	0.016	0.112		0.108	0.369	0.626	0.100
**Per4**	−0.003	0.124	0.025	0.056	0.070	−0.002	0.031	0.377	0.082	1	0.373	0.481	0.242
	0.961	0.015	0.63	0.277	0.169	0.962	0.546	<0.001	0.108		<0.001	<0.001	<0.001
**Per7**	−0.102	0.009	−0.066	−0.056	−0.066	−0.045	−0.016	0.265	0.046	0.373	1	0.653	0.270
	0.045	0.867	0.196	0.277	0.195	0.38	0.749	<0.001	0.369	<0.001		<0.001	<0.001
**Per10**	−0.094	−0.025	−0.101	−0.061	−0.072	−0.104	−0.018	0.451	−0.025	0.481	0.653	1	0.286
	0.067	0.630	0.048	0.234	0.161	0.041	0.725	<0.001	0.626	<0.001	<0.001		<0.001
**Per13**	0.091	0.133	0.124	0.198	0.140	0.069	0.096	0.123	0.084	0.242	0.270	0.286	1
	0.076	0.009	0.015	<0.001	0.006	0.179	0.060	0.016	0.100	<0.001	<0.001	<0.001	

**Table 4 ijerph-19-08474-t004:** Pearson correlations between the Grit-S factors, the original factors of the Motivational Persistence scale and the factors adapted to the Spanish population.

	RUP	CPP	LTPP	RUPs	PPs
Consistency	−0.287 **	0.329 **	0.372 **	−0.314 **	0.375 **
Perseverance	−0.029	0.562 **	0.485 **	−0.056	0.563 **

RUP: Recurrence of unattained purposes; CPP: Current purposes pursuing; LTPP: Long-term purposes pursuing; RUPs: Recurrence of unattained purposes of spanish version; PPs: Purpose Pursuing of spanish version). ** *p* < 0.001.

**Table 5 ijerph-19-08474-t005:** Goodness-of-fit indices of invariance models by age group (≤20 and >20 years) and sex.

Invariance	χ^2^	gl	χ^2^/gl	RMSEA (CI at 90%)	IFC	AIC
**Age Group**						
≤20 years	60.7		1.41	0.044 (0.011–0.068)	0.972	128.8
>20 years	60.8		1.41	0.049 (0.012–0.077)	0.964	128.7
Configural	121.4	86	1.41	0.033 (0.018–0.046)	0.966	257.4
Metrics	139.6		1.47	0.035 (0.022–0.047)	0.948	257.6
Climb	155.3		1.47	0.035 (0.022–0.046)	0.952	251.3
Strict	172.5		1.47	0.035 (0.023–0.046)	0.946	246.5
**Sex**						
Men	72.9		1.69	0.048 (0.028–0.067)	0.960	140.9
Women	36.3		0.85	0.001 (0.001–0.053)	0.999	104.3
Configural	109.3	86	1.27	0.027 (0.004–0.041)	0.977	245.3
Metrics	120.5		1.27	0.026 (0.006–0.040)	0.975	238.5
Climb	142.5		1.34	0.030 (0.015–0.042)	0.964	238.1
Partial Scalar	139.7		1.33	0.029 (0.014–0.042)	0.966	237.2
Strict	167.5		1.43	0.035 (0.023–0.046)	0.950	241.5
Strict Partial	159.3		1.39	0.032 (0.019–0.043)	0.956	237.3

**Table 6 ijerph-19-08474-t006:** Mean ± standard deviation of the weekly hours of Deliberate Practice (DP) and of the factors of the Motivational Persistence scale (RUP: Recurrence of unattained purposes and PP: Purposes pursuing) as a function of competitive level and category.

	Cadets		Junior	
	≤1500 Ranking	≥5000 Ranking	≤1500 Ranking	≥5000 Ranking
	(N = 42)	(N = 40)	(N = 20)	(N = 11)
DP				
Woman	9.48 ± 3.54	3.25 ± 0.61	9.83 ± 4.17	3.00 ± 0.01
Male	10.77 ± 4.03	5.29 ± 3.46	13.14 ± 5.07	4.17 ± 1.03
RUP				
Woman	2.68 ± 0.61	3.13 ± 0.41	3.04 ± 0.86	3.38 ± 0.18
Male	2.77 ± 0.93	3.13 ± 0.74	2.70 ± 0.61	2.94 ± 0.79
PP				
Woman	3.84 ± 0.75	3.19 ± 0.52	3.43 ± 0.48	3.43 ± 0.20
Male	3.92 ± 0.68	3.77 ± 0.63	3.86 ± 0.74	3.75 ± 0.38

Note. Cadets: ≤1500 ranking (22 males, 20 females); ≥5000 ranking (34 males 6 females); Juniors: ≤1500 ranking (14 males, 6 females); ≥5000 ranking (9 males, 2 females).

## Data Availability

The data presented in this study are available on request from the corresponding author. The data are not publicly available due to restrictions of privacy.

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
