# Peer review of "Achievement Goals across Persistence—Validation of the Spanish Version of the Motivational Persistence Scale"

_ijerph, 2022, doi:10.3390/ijerph19148474_

Round 1

Reviewer 1 Report

Thank you for the opportunity to review the paper. 

Although the topic is interesting, the introduction does not support the aims of the study. There is no clear justification for choosing the questionnaire and there is no hypothesis. There are various errors in citations and wording (line 41, 51, 53 , etc.). Results need to be more clearly presented. I suggest splitting it into two different papers so that you can deepen the results and discuss each objective. 

Author Response

Thank you so much for your comments. I have tried to do all my best for answering your questions

The introduction does not support the aims of the study. There is no clear justification for choosing the questionnaire and there is no hypothesis.

Response

Introduction. I have included and hypothesis and I have included articles to justificate the topic.

There are various errors in citations and wording (line 41, 51, 53 , etc.)

Response

I have reviewed the document the references are done across the journal citation https://res.mdpi.com/data/mdpi_references_chicago_guide-update-v6.pdf

Results need to be more clearly presented. I suggest splitting it into two different papers so that you can deepen the results and discuss each objective. 

Response

We have decided to make only a complete article of two studies because, in this case, we can explain all validation progress.

Reviewer 2 Report

In my opinion, this is not a strictly scientific work, however in the future it could be used for both scientific and practical purposes.  I think that results of your work is more important for Spanish-language readers. Therefore they should be published in Spanish-language journals. Besides, I have some doubts and questions in relation to the text. Did you have permission from the Bioethics Committee to conduct the study, and in the case of adolescents, parental consent.

I also have a question why you divided the study group by age into two subgroups: younger than 20 and older than 20. What justifies such a division if the upper age limit in the study group was 46 years.   You did not provide the mean age of the   subgroups, we do not know the age distribution. 

Why you divided the particpants according to ranking into such different groups (those with a ranking of 1500th place or better versus those ranked 5000th place or worse) - no justification

Minor remarks

Abbreviations used for the first time in the text should be explained - it is not enough that the expansions  are included in the Abstract

l.133 The authors probably meant:  coefficient alpha (Cronbach's alpha), not "cronbrach alpha information"

l. 327 whether the abbreviation PD means the same as DP?

l. 378 what does "39" mean in this line, is it a reference number (no brackets)?

Author Response

Did you have permission from the Bioethics Committee to conduct the study, and in the case of adolescents, parental consent.

Response

I have included Informed consent was obtained from all older subjects and parents of younger involved in the study.

In study 2 I have included: After parents of players read and signed the informed consent form approved by the university’s Institutional Board, they were able to continue to the questionnaires, if they consented.

Abbreviations used for the first time in the text should be explained - it is not enough that the expansions  are included in the Abstract

Response

I have explain the abreviations inside the text. Deliberate Practice (DP), Purpose Pursuing (PP) and Recurrence of Unattained Purposes (RUP).

l.133 The authors probably meant:  coefficient alpha (Cronbach's alpha), not "cronbrach alpha information"

Response

It has been changed L 149

l. 327 whether the abbreviation PD means the same as DP?

Response

It has been changed L 370

l. 378 what does "39" mean in this line, is it a reference number (no brackets)?

It has been changed L 389

Reviewer 3 Report

The article is methodologically consistent and it shows a refinement in the topic analyzed. The only section to improve is actualizing the bibliography.

Author Response

Thank you so much for your comments.

The only section to improve is actualizing the bibliography.

Response

I have include studies from 2019, 2020, 2022. See references.

Reviewer 4 Report

Review of the manuscript entitled: Achievement Goals across Persistence. Validation of the Spanish Version of the Motivational Persistence Scale of Constantin et al. 2011. The manuscript submitted is appropriate to the subject matter and scientific rigor. The authors raised a very current issue at work, which is not only interesting from a scientific but also a practical point of view. Some remarks improving the quality of future research. and suggested changes and comments to the submitted manuscript in order to improve the quality of the planned research and future publications below:

1.      In the discussion, the authors only refer to a few articles from 1993-2012. [36;37;38;40;6;26]. May be authors should refer to more publications and more recent research (2019-2022). Are they sure there aren't any, no one is conducting similar research? ? May I be reviewing the literature again? And refer to them in the discussion ???? Has nobody tried to validate this scale so far ??

2.      Would you please correct the bibliography in accordance with the journal's guidelines and standardize it. Put the full namne of Jurnal or abbreviations, pages numbers, DOI numbers., Would you please add the acess for the article. Some of the cited items have the wrong access address and some mistakes in the title in addition, there is a symbol "y" there instead of „and” or „&” correct it, please See my comments below.

3.      Would you please add Table 4 in this article. I don't see table 4 in this article (but the authors describe the results in the text).

4.      Maybe the data in Figures 1-4 should be described in a different way, because I do not see in this figures the data provided by the authors in the text describing their results.

5.      Would you please cheek it and correct : „... Demir and Doner [39] found a positive correlation between RUP and PP with job performance among Teachers.” In the article[{39] as I see, candidates for teachers were surveyed, not teachers.

Remaining comments in the article is below.

King regards,

Author Response

Thank you so much for your recommendations. I have tried to answer all of them.

 In the discussion, the authors only refer to a few articles from 1993-2012. [36;37;38;40;6;26]. May be authors should refer to more publications and more recent research (2019-2022). Are they sure there aren't any, no one is conducting similar research? ? May I be reviewing the literature again? And refer to them in the discussion ???? Has nobody tried to validate this scale so far ??

Response

I have include a few articles from 2022, 2019 and 2020.( see 10, 11, 12) The have studied this topic in student as well. I have include a turkish validation (37) as well to justificate our results.

Would you please correct the bibliography in accordance with the journal's guidelines and standardize it. Put the full namne of Jurnal or abbreviations, pages numbers, DOI numbers., Would you please add the acess for the article. Some of the cited items have the wrong access address and some mistakes in the title in addition, there is a symbol "y" there instead of „and” or „&” correct it, please See my comments below.

Response

Following your document, i have change all the references i have write them following this document https://mdpi-res.com/data/mdpi_references_guide_v5.pdf.

In addition i have checked all the articles that you told me and I have eliminated one. Scanlan, T. K., Carpenter, P. J., Simons, J. P., Schmidt, G. W., & Keeler, B. (1993). An Introduction to the Sport Commitment 414
Model. Journal of Sport and Exercise Psychology, 15(1), 1-15. https://doi.org/10.1123/jsep.15.1.1.

 Would you please add Table 4 in this article. I don't see table 4 in this article (but the authors describe the results in the text).

Response

Table 4 is included.

Would you please cheek it and correct : „... Demir and Doner [39] found a positive correlation between RUP and PP with job performance among Teachers.” In the article[{39] as I see, candidates for teachers were surveyed, not teachers

It have been changed by candidates